# Effect of Amino Acids on *Fusarium oxysporum* Growth and Pathogenicity Regulated by TORC1-*Tap42* Gene and Related Interaction Protein Analysis

**DOI:** 10.3390/foods12091829

**Published:** 2023-04-28

**Authors:** Yijia Deng, Rundong Wang, Yuhao Zhang, Jianrong Li, Ravi Gooneratne

**Affiliations:** 1College of Food Science, Southwest University, Chongqing 400715, China; ikea7713@163.com; 2College of Food Science and Engineering, Bohai University, Jinzhou 121013, China; lijr6491@163.com; 3Chongqing Key Laboratory of Speciality Food Co-Built by Sichuan and Chongqing, Chongqing 400715, China; 4Department of Wine, Food and Molecular Biosciences, Faculty of Agriculture and Life Sciences, Lincoln University, Lincoln 7647, New Zealand

**Keywords:** amino acids, *Fusarium oxysporum*, T-2 toxin, *Tap42*, KEGG

## Abstract

Free amino acids (AAs) formed in fermented meat products are important nitrogen sources for the survival and metabolism of contaminating fungi. These AAs are mainly regulated by the TORC1-*Tap42* signaling pathway. *Fusarium* spp., a common contaminant of fermented products, is a potential threat to food safety. Therefore, there is an urgent need to clarify the effect of different AAs on *Fusarium* spp. growth and metabolism. This study investigated the effect of 18 AAs on *Fusarium oxysporum* (Fo17) growth, sporulation, T-2 toxin (T-2) synthesis and *Tri5* expression through *Tap42* gene regulation. Co-immunoprecipitation and Q Exactive LC-MS/MS methods were used to detect the interacting protein of Tap42 during specific AA treatment. *Tap42* positively regulated L-His, L-Ile and L-Tyr absorption for Fo17 colony growth. Acidic (L-Asp, L-Glu) and sulfur-containing (L-Cys, L-Met) AAs significantly inhibited the Fo17 growth which was not regulated by *Tap42*. The L-Ile and L-Pro addition significantly activated the sporulation of ΔFo*Tap42.* L-His and L-Ser inhibited the sporulation of ΔFo*Tap42*. In T-2 synthesis, ΔFo*Tap42* was increased in GYM medium, but was markedly inhibited in L-Asp and L-Glu addition groups. Dose–response experiments showed that 10–70 mg/mL of neutral AA (L-Thr) and alkaline AA (L-His) significantly increased the T-2 production and *Tri5* expression of Fo17, but *Tri5* expression was not activated in ΔFo*Tap42*. Inhibition of T-2 synthesis and *Tri5* expression were observed in Fo17 following the addition of 30–70 mg/mL L-Asp. KEGG enrichment pathway analysis demonstrated that interacting proteins of Tap42 were from glycerophospholipid metabolism, pentose phosphate pathway, glyoxylate and dicarboxylate metabolism, glycolysis and gluconeogenesis, and were related to the MAPK and Hippo signaling pathways. This study enhanced our understanding of AA regulation in fermented foods and its effect on *Fusarium* growth and metabolism, and provided insight into potential ways to control fungal contamination in high-protein fermented foods.

## 1. Introduction

Fermented meat (sausage, bacon, ham) and fermented dried aquatic (dried/smoked fish, dried shrimp) products are two types of flavor foods, processed under natural or control drying conditions. Flavor formation occurs through a series of biochemical and physical changes and depends on microorganism fermentation and/or endogenous enzyme action [1,2,3]. During fermentation, a number of aromatic substances, such as free amino acids (AAs), aldehydes, alcohols, acids, heterocyclic compounds and nucleotides, can be formed which impart a unique flavor to the food [4,5,6,7]. Due to lower water activity (aw) and higher salinity, fermented dry/semi-dry products can be stored for a long time with minimal microbial reproduction. However, unpackaged fermented products are at risk of contamination by environmental and suspended microorganisms in the air, including fungi, which adhere easily to the product surface. During long-term storage, protein breakdown in food forms free AAs to provide nutrients for fungal growth and metabolism. In addition, some fungal contaminants such as *Penicillium* spp., *Aspergillus* spp., *Fusarium* spp. and *Candida* spp. can utilize AAs to grow at aw < 0.9 [8,9] and contaminate fermented food [10,11]. Among the common contaminated fungi, *Fusarium* spp. are plant pathogenic fungi which generally infect high-carbon crops such as grain, corn, soybean and wheat to produce mycotoxins such as T-2 toxin (T-2), HT-2 toxin (HT-2), deoxynivalenol (DON) and 15-acetyl deoxynivalenol (15Ac-DON) [12,13,14]. Among these, T-2 is highly toxic and causes weight loss [15], neurological disorders [16], immunosuppression [17], bone marrow damage [18] and cutaneous toxicity [19]. The *Tri5* gene plays an important role as the initiator of T-2 synthesis [20]. Several studies have reported that *Fusarium* contamination in dried/smoked fish and fermented meat products sold at markets poses a potential threat to consumer health [10,21,22,23,24]. Proteins in meat products are degraded to AAs during long-term storage due to microbial growth. Rabie et al. (2014) reported a reduction in free AA concentration in horse and beef sausage during 28-day storage period due to contaminating microorganism metabolism [25]. Although protein degradation into free AAs such as aspartic acid (L-Asp), glutamic acid (L-Glu) and alanine (L-Ala) can enhance the unique flavor of fermented products, long storage periods can lead to excessive proteolysis, resulting in the accumulation of free AAs, which provide conducive conditions for *Fusarium* growth and the production of spores and mycotoxins. Furthermore, fermented meat products can degrade into more than 22 AAs, including threonine, serine, glutamine, lysine, tyrosine and histidine. These AAs can be classified into acidic-, alkaline-, neutral and sulfur-containing AAs. Regulatory mechanism of each AA on fungal metabolism vary between species. At present, few studies have reported the effects of amino acid on *Fusarium* sp. growth and metabolism [26]. To understand the metabolic mechanism of *Fusarium* sp. on AAs, it is important to determine how fungi utilize different nutrients in high-protein fermented foods.

The target of rapamycin (TOR) protein is a key regulator of eukaryotic growth and a class of evolutionarily conserved serine/threonine (Ser/Thr) protein kinase [27]. The TORC1 signaling pathway includes two parts: (1) upstream activation pathway of TORC1, consisting of GTPases Gln and Gtr [28] regulated by *Gln3* [29] and *Gtr1*/*Gtr2* [30], respectively, allowing fungi to respond to AAs, stress and other environmental signals; (2) two branches of the downstream pathway, controlled by *Sch9* and *Tap42,* which regulate fungal growth and metabolism, respectively [31], including protein translation, ribosomal synthesis, gene transcription, protein degradation and autophagy [32]. Most studies on the regulation of AAs on fungal growth and metabolism have examined the TORC1 signaling pathway [33].

A complete response system was investigated in *Saccharomyces cerevisiae* under a limited nitrogen condition [34]. *Tap42*-dependent phosphorylation occurs when the TOR signal is activated, promoting the interaction between *Tap42* and phosphatase. When the external nutrient conditions are unfavorable, phosphatase isolated from Tap42 and many downstream targets of the TOR signaling pathway, such as *Npr1*, *Ure2* and *Gln3,* are dephosphorylated to regulate metabolism [35]. Therefore, *Tap42* is an important metabolic regulator used by fungi to respond to external AAs and control *Fusarium* metabolism.

The aim of this study was to elucidate the role of 18 AAs as nitrogen sources for *Fusarium* spp. growth (colony morphology) and metabolism (sporulation, toxin production, *Tr5* expression) regulated by the *Tap42* gene. Co-immunoprecipitation was used to capture the interacting proteins of Tap42 following exposure to specific AAs. This provides an experimental basis for controlling fungal contamination in fermented meat products from an AA perspective.

## 2. Materials and Methods

### 2.1. Chemicals and Experimental Strains

Biological reagents (≥99%) L-Aspartic acid (L-Asp), L-Glutamic acid (L-Glu), L-Serine (L-Ser), Glycine (Gly), L-Histidine (L-His), L-Threonine (L-Thr), L-Arginine (L-Arg), L-Proline (L-Pro), L-Alanine (L-Ala), L-Tyrosine (L-Tyr), L-Valine (L-Val), L-Methionine (L-Met), L-Cysteine (L-Cys), L-Isoleucine (L-Ile), L-Leucine (L-Leu), L-Tryptophan (L-Try), L-Phenylalanine (L-Phe) and L-Lysine (L-Lys) were purchased from Ruiyong Biological Technology (Shanghai, China). T-2 toxin standard was purchased from Enzo Life Science (Farmingdale, NY, USA). Chromatographic-grade reagents (≥99%) methanol, ethyl acetate, acetonitrile and formic acid were purchased from Sigma-Aldrich (Shanghai, China). Kanamycin, dithiothreitol (DTT), isopropyl-beta-D-thiogalactopyranoside (IPTG), and 1-(p-Toluenesulfonyl) imidazole (C_10_H_10_N_2_O_2_ S) were purchased from Xiya Chemical Technology (Qingdao, China). IP lysis, 1× loading buffer and trypsin were obtained from Sangon Biotech (Shanghai, China). Disodium hydrogen phosphate, sodium chloride, iodoacetamide, ammonium bicarbonate, ammonium dihydrogen phosphate, potassium chloride, magnesium sulfate heptahydrate, dipotassium hydrogen phosphate, ferrous sulfate heptahydrate and ammonium bicarbonate were purchased from Xilong Science (Shantou, China).

Wild-type *Fusarium oxysporum* (Fo17, GDMCC 60824, isolated from dried fish) has a strong T-2 toxin synthesis ability. Gene deletion strain ΔFo*Tap42* and complement strain ΔFo*Tap42*-C were obtained by the double-crossover homologous recombination method (in a preliminary experiment). Top10 *Escherichia coli* and Rossetta (DE3) competent cells were purchased from Kamede Biological (Tianjin, China).

### 2.2. Colony Growth Analysis

A dextrose agar medium (20 g glucose, 20 g agar 0.1 g chloramphenicol) containing 5 mg/mL each of 18 AAs was prepared. For the control group, PDA medium was used without AA addition. The wild-type Fo17, ΔFoT*ap42* and ΔFoT*ap42*-C were inoculated into a AA dextrose agar medium with a 5 mm mycelium disk, cultured for 7 d at 28 °C. The colony’s morphology was observed and photographed.

### 2.3. Sporulation Capacity Analysis

A 1L Czapek dox agar (CDA) liquid medium (nitrogen (N) source removed, containing 1 g K_2_HPO_4_, 0.5 g KCl, 0.5 g MgSO_4_·7H_2_O, 0.01 g FeSO_4_·7H_2_O, 30 g saccharose) containing 10 mg/mL each of 18 AAs were prepared. The control group used a CDA liquid medium with an original nitrogen source and without AA addition. The wild-type Fo17, ΔFoT*ap42* and ΔFoT*ap42*-C were inoculated into the CDA medium with a 5 mm mycelium disk, followed by shaking at 120 rpm/min at 28 °C for 7 d. Next, spores in cultured solution were dispersed with a 84-1A magnetic agitator (Sile, Shanghai, China) at a speed of 500 rpm/min and filtered with three layers of gauze. The 50 μL filtrate was transferred to a blood cell-counting plate to calculate the number of spores using a CX23 optical microscope (Puch, Shanghai, China).

### 2.4. T-2 Toxin-Producing Ability Analysis

A 1L GYM liquid medium (N source removed, containing 0.2 g KCl, 0.2 g MgSO_4_·7H_2_O, 10 g glucose, 5 g yeast extract, 1 mL 0.005 g/L CuSO_4_·5H_2_O, 1 mL 0.01 g/L ZnSO_4_·7H_2_O) containing 5 mg/mL each of 18 AAs were prepared. The control group used a GYM liquid medium with original N source and without AA addition. A 1 mL GYM solution was added into a 2 mL centrifuge tube. Next, wild-type Fo17, ΔFo*Tap42* and ΔFo*Tap42*-C were inoculated into the GYM medium with two 5 mm mycelium disks and cultured at 28 °C for 14 d. After culturing, the solution was centrifuged (Xiangyi, Changsha, China) at 5000 rpm for 10 min, the supernatant was collected, 1 mL ethyl acetate was added and the solution was mixed using a XW-80A vortex (Fudi, Fuzhou, China) for 5 min. The supernatant was collected and dried at 60 °C in N. Next, 1 mL 30% methanol was added to redissolve it; then, it was filtered with a 0.22 μm filter and T-2 was detected by LC-MS/MS.

Toxin analysis was performed on a Thermo Scientific Surveyor HPLC (Thermofisher, Waltham, MA, USA)system which comprised a Surveyor MS Pump Plus, an on-line degasser and a Surveyor Autosampler Plus coupled with a Thermo TSQ Quantum Access tandem mass spectrometer equipped with an electrospray ionization (ESI) source (Thermofisher, Waltham, MA, USA). The separation was performed at 35 °C using a Hypersil GOLD column (5 μm, 100 mm × 2.1 mm) (Thermofisher, Waltham, MA, USA) at a flow rate of 0.25 mL/min. The mobile phase consisted of methanol (A) and water containing 5 mM ammonium acetate 0.1% formic acid (B), with a gradient elution program as follows: 0 min 30% A, 3.0 min 90% A, 5 min 90% A and 3 min 30% MS/MS detection was carried out using a triple quadruple mass spectrometer coupled with an electrospray ionization source operating in positive (ESI+) mode (Shimadzu, Kyoto, Japan). The ionization source parameters were set as follows: spray voltage, 4500 V; sheath gas pressure, 35 au; ion sweep gas pressure, 0 au; auxiliary gas pressure, 15 au; capillary temperature, 350 °C; tube lens offset, 118 V; skimmer offset, 0; collision energy, 1.5 eV; and collision pressure, 1.5 mTorr.

### 2.5. T-2 Synthesis and Tri5 Expression Dose-Response Relationship Analysis

The 50 mL GYM liquid medium (N source removed) containing 10, 30, 50 or 70 mg/mL of L-Thr, L-His and L-Asp was prepared, and the *F. oxysporum* Fo17, ΔFo*Tap42* and ΔFo*Tap42*-C were inoculated into a 1 mL GYM medium with a 5 mm mycelium disk. Next, the samples were cultured at 28 °C and centrifuged at 120 rpm/min for 14 days. Then, the culture solution was treated as explained in Section 2.4, and T-2 was detected by LC-MS/MS.

The 25–50 mg (dry weight) cultured mycelia of Fo17, ΔFo*Tap42* and ΔFo*Tap42*-C were weighed and ground into powder with liquid N. Total RNA was extracted using a Spin Column Fungal Total RNA Purification Kit (Sangon Biotech, Shanghai, China). Purity and concentration were determined by a nucleic acid quantitative analyzer (Thermofisher, Waltham, MA, USA). A StarScript II First-strand cDNA Synthesis Kit (Genstar, Beijing, China) was used to transcribe RNA into cDNA. The HATri/F (5-CAGATGGAGAACtGGATGGT-3) and HATri/R (5-GCACAAGTGCCACGTGAC-3) were used as primer pairs, and the β-Tubulin/F (TTCCCCCGTCTCCACTTCTT) and β-Tubulin/R (GACGAGATCGTTCATGTTG) were used as the internal control gene primers. The reaction system of the qRT-PCR mixture was prepared as a 10 μL reaction (containing 5 μL 2× SYBR Green Supermix, 0.4 μL of each pair of primers (10 μM), 0.2 μL ROX Reference Dye, 1.0 μL cDNA and 3.0 μL PCR-certified water). The amplification reaction cycling procedure was as follows: 5 min at 95 °C, followed by 40 PCR cycles of 10 s at 95 °C for denaturation and 30 s at 60 °C for annealing and elongation. The melting curve analysis consisted of 3 s at 60 °C, followed by heating up to 95 °C with a ramp rate of 1 °C/3 s. If the melting curves showed a clear single peak, it meant the primers were specific. The experiment was conducted for each sample in triplicate. Based on the PCR reaction’s CT value, *Tri5* expression was analyzed by the 2^−ΔΔCt^ method.

### 2.6. TORC1-Tap42 Interacting Protein Analysis

#### 2.6.1. Construction of Tap42 Expression Vector

Based on the NCBI website (https://www.ncbi.nlm.nih.gov/, accessed on 27 May 2022) for the Tap42 sequence and related protein sequence, analyzed the protein signal peptide to obtain the corresponding base sequence. The Tap42 template was synthesized by chemical synthesis and cloned into a pET-28a(+) (3228 bp) expression vector by the BamHI-XhoI double enzyme digestion method. The constructed expression vector was transformed into a Top10 *E. coli* clone, coated on an agar plate (containing 30 μg/mL kanamycin) and cultured overnight at 37 °C. A single cloned strain with the correct sequence was selected. A 100 μL sample of Rossetta (DE3) competent cells was mixed with 10 μL pET28a(+)-TAP42 plasmid, bathed in ice for 30 min, quickly heated at 42 °C for 90 s and placed in an ice bath for 5 min. The solution was coated on LB medium (containing 50 μg/mL kanamycin) and cultured at 37 °C for 14 h. Monoclones were selected for the induction culture. The cloned strain was cultured at 37 °C for 200 mL until the OD 600 was 0.6–0.8. IPTG was added to the medium for induction at 16 °C for 24 h and centrifuged at 5000 rpm for 5 min. The supernatant was filtered by a 0.22 μm filter and then enriched by Ni column affinity chromatography.

#### 2.6.2. Protein Purification by Ni Column Affinity Chromatography

A binding buffer (0.02 M Na_2_HPO_4_, 0.5 M NaCl, pH = 7.4) was used to balance the Ni column, with a flow rate of 3 mL/min. The supernatant, collected in Section 2.6.1, was purified using a AKTA purifier (Explorer 10, Cytiva, Sweden). After sample loading, in order to remove the impurities, the Ni column was washed with buffer (0.02 M NaH_2_PO_4_, pH = 7.4, 0.5 M NaCl, 20 mM 1-(p-Toluenesulfonyl) imidazole) until the UV signal returned to the baseline. Elusion buffer (0.02 M Na_2_HPO_4_, 0.5 M NaCl, 500 mM 1-(p-Toluenesulfonyl)imidazole, pH = 7.4) was used to wash the Ni column until the sample was eluted. The samples were collected for SDS-PAGE expression.

#### 2.6.3. Co-Immunoprecipitation Analysis

(1) Sample treatment: The cultured mycelia were washed with precooled 0.01 M PBS 3 times and centrifuged at 12,000 rpm for 1 min each time to collect the mycelia. Liquid N was used to grind it into a powder; then, it was redissolved in 1 mL IP lysis, centrifuged at 12,000 rpm for 2 min to collect the supernatant and, finally, collected on ice. (2) Next, we prepared 100 μL Ni beads in triplicate, centrifuged them at 12,000 rpm and 4 °C for 1 min and removed the supernatant. Then, we added 800 μL IP lysate, centrifuged it at 12,000 rpm and 4 °C for 1 min and removed the supernatant. This step was repeated 3 times, and the final supernatant was resuspended in 200 μL IP lysate. (3) The 10 μg of Tap42 protein was prepared by adding it into the solution and incubating at 4 °C for 60 min, followed by dividing the solution into one tube. The sample supernatant prepared in step (1) was added to the tube, incubated at 4 °C for 1 h and centrifuged at 2000 rpm at 4 °C for 1 min to remove the supernatant. Next, 1.6 mL IP lysate was added and centrifuged at 2000 rpm 4 °C for 1 min to remove the supernatant. This step was repeated five times. (4) A 30 μL sample of 1× loading buffer was added into the sample tube and boiled at 100 °C for 10 min. Next, the sample was stored at −80 °C for further studies.

#### 2.6.4. Sodium Dodecyl Sulfate-Polyacrylamide Gel Electrophoresis (SDS-PAGE)

The samples (from procedure Section 2.6.3) were defrosted and centrifuged at 12,000 rpm at 4 °C for 10 min. The supernatant was mixed with one volume of load buffer (187.5 mM Tris (pH 6.8), 6% SDS, 30% glycerol and 15% β-mercaptoethanol) and heated at 98 °C for 5 min to completely denature the proteins (as sample solutions for SDS-PAGE). Polyacrylamide gels (6%) were prepared in Bio-Rad chambers (separation gel: 2.5 μL (acrylamide 30%, bisacrylamide 0.8%), 1.5 μL Tris-HCl (1.5 M pH 8.8), 52.5 μL 10% SDS, 955 μL distilled water, 150 μL PSA, and 7.5 μL TEMED; stacking gel: 312 μL (acrylamide 30%, bisacrylamide 0.8%), 450 μL Tris-HCl (0.5 M pH 6.8), 1.0 μL distilled water, 50 μL PSA and 4.0 μL TEMED). The electrophoresis chamber was filled with running buffer (1.44% glycine, 0.3% Tris, and 0.1% SDS). Next, the sample solutions (10 μL) and 10 μL of protein marker were loaded onto the gel. Gel electrophoresis was performed at a constant voltage of 50 V until the sample reached the separation gel, at which point the voltage was increased to 100 V. The gels were stained with a Protein Stain Q Kit (Sangon Biotech, Shanghai, China). Gel images were processed using the Image lab Software (version 6.01, BioRad, Hercules, CA, USA), adjusting the gamma setting to improve the contrast.

#### 2.6.5. LC-MS/MS Analysis of Tap42 Interacting Proteins

The electrophoretic target band was cut, transferred into a 50 mL centrifuge tube and rinsed twice with ultrapure water; then, a mixture of 25 mM NH_4_HCO_3_ and 50% acetonitrile was added to decolorize for 30 min. To the extracted discolored solution, a dehydrated solution 1 of 50% acetonitrile was added and let stand for 30 min; then, it was sucked out and the dehydrated solution 2 of absolute ethyl alcohol was added and let stand for 30 min. The target band was freeze-dried in a vacuum (Xiangyi, Changsha, China) at −20 °C for 2 h to obtain a lyophilized gel block, 50 μL reductive solution 1 was added (10 mM DTT, 25 mM NH_4_HCO_3_) and it was left to stand in a bath at 57 °C for 1 h. Next, the solution was removed, 50 μL reductive solution 2 was added (50 mm iodoacetamide, 25 mM NH_4_HCO_3_), it was placed at room temperature (RT) for 30 min and then the solution was removed. For 10 min, 10% ethyl alcohol was added and the solution was removed. Then, the dehydrated solution 1 was added for 30 min, it was removed and the dehydrated solution 2 was added for 30 min. The dehydrating solution 2 was removed and 10 μL enzymatic hydrolysate was added (25 mM NH_4_HCO_3_ containing 0.02 μg/μL trypsin) to hydrolyze overnight at 37 °C. Next, the solution was centrifuged at 12,000 rpm for 2 min to obtain the supernatant for protein analysis. A Thermofisher QE Orbitrap high field electrostatic field orbital trap mass spectrometer (Thermofisher, Waltham, MA, USA) was used to detect the interacting proteins. The first-order spectrum scanning range was 350–1600 *m*/*z*; the second-order mass spectrometry mode was CID (Elite). A new protein library was constructed using the theoretical amylase sequence. The main search parameters were as follows: (1) immobilization modification: carbomimomethyl on Cys; (2) variable modification: oxidation on Met, acetylation on protein N-terminal; (3) allowable error of parent ion of peptide: 10 ppm; (4) allowable error of fragment ion: 0.6 Da.

### 2.7. Statistical Analysis

All experiments were conducted three times in parallel. The data were expressed as mean ± standard deviation (SD) and statistically analyzed by IBM SPSS statistics software (Version 26.0, BioRad, Hercules, CA, USA). Significant differences between the control and the treated fish were determined by one-way analysis of variance (ANOVA), followed by the Tukey’s test to compare the control and treatment group values. A *p*-value of <0.05 was considered significant.

## 3. Results and Discussion

### 3.1. Effects of AAs on Colony Growth of F. oxysporum Regulated by Tap42

Rapamycin-sensitive TORC1 protein kinase is an important component of a conserved signal-cascading mechanism which controls cellular absorption and the response of AAs to regulate fungal growth, metabolism and pathogenicity [36]. The *Tap42* gene is the key regulator to absorb and metabolize AAs for fungi. The effects of AAs on the colony morphology of Fo17, ΔFo*Tap42* and ΔFo*Tap42*-C are shown in Figure 1. On PDA medium, Fo17 grew normally with several aerial hyphae, while ΔFo*Tap42* grew relatively slowly with fewer aerial hyphae. Colony morphology recovered to normal conditions following *Tap42* supplementation. The L-His, L-Leu, L-Pro, L-Thr and L-Tyr additions as nitrogen sources significantly activated the growth of Fo17, with larger hyphal diameters but fewer mycelia in the 7-d culture. The colony was significantly smaller in ΔFo*Tap42*, especially in the L-Try and L-Tyr media. This may be because the *Tap42* target-regulated the absorption of L-Try and L-Tyr, while a lack of *Tap42* decreased the response of the TORC1 pathway, which resulted in growth inhibition. In the L-Val medium, the colony of ΔFo*Tap42* was smaller, with radial mycelia, but recovered to the normal level in ΔFo*Tap42*-C. Sulfur-containing (L-Cys, L-Met) and acidic (L-Asp, L-Glu) AAs significantly inhibited the growth of Fo17, ΔFo*Tap42* and ΔFo*Tap42*-C, which indicated that the absorption of these two AAs was not regulated by *Tap42*. Supplementation with L-Cys and L-Met resulted in less mycelial growth, while L-Asp and L-Glu caused thicker hemispherical mycelia. These results are in accordance with Shiobara et al. [26], who reported an inhibitory effect of L-Met and L-Cys on *Fusarium grainei* growth. Hence, S-containing AAs can be used as mycotoxin antidotes [37] and fungal inhibitors [38] because the sulfhydryl groups prevent normal fungal growth and metabolism. The L-Asp and L-Glu were not conducive to Fo17 and ΔFo*Tap42* growth, possibly because acidic AAs significantly decreased the pH of the medium. *F. oxysporum* Fo17 prefers growth at pH > 6 (unpublished data).

### 3.2. Effect of AAs on Spore Production and T-2 Synthesis of F. oxysporum Regulated by Tap42

The spore production and T-2 synthesis of *F. oxysporum* Fo17, ΔFo*Tap42* and ΔFo*Tap42*-C were significantly different and regulated by AAs (Table 1). Compared to the control group (CDA medium), the spore production of Fo17 in the L-Arg, L-Lys, L-Met, L-Phe and L-Pro groups were significantly elevated (*p* < 0.05), especially in the L-Lys group (22.30 × 10^5^ CFU/L, *p* < 0.05), but were decreased in the L-Asp, L-Cys, L-Gly, L-Leu, L-Thr, L-Try and L-Val groups, with values ranging from 3.14–7.50 × 10^5^ CFU/L. Following *Tap42* deletion, the spore production of ΔFo*Tap42* significantly decreased (*p* < 0.05) in most groups with added AAs. In contrast, L-Arg, L-Ile, L-Lys, L-Met, L-Phe and L-Pro addition significantly increased (*p* < 0.05) the spore production of Fo17. This was most evident in the L-Ile- and L-Pro-added groups, especially the ΔFo*Tap42,* with values ranging from 35.80–36.26 × 10^5^ CFU/L. Non-polar AAs such as L-Ile and L-Pro are generally hydrophobic, which is not conducive for the absorption of pathomycetes such as *Fusarium* sp.; thus, the speed of fungal growth and spore production is limited [39]. The spore production of *Fusarium* sp. Was positively regulated by *Tap42* in both the control and most AA-added groups, except L-Asp, Gly, L-Thr and L-Val. Alfatah et al. [40] reported that, in a nutrient-dependent developmental state, *Tap42-Sit4-Rrd1*/*Rrd2* signaled for the regulation of spore germination, while *Tap42* deletion inhibited the TORC1 pathway and blocked spore germination. Therefore, a lack of *Tap42* leads to inactivation of a part of TORC1 function, as well as an inability to respond to glucose and other exogenous AAs, resulting in reduced sporulation.

The T-2 toxin production of Fo17 was 22.86 ng/mL in the control group (GYM medium), which increased in ΔFo*Tap42* (30.82 ng/mL). When supplemented with *Tap42*, the T-2 production returned to normal. Contrary to the spore production capacity, this result indicated that a lack of the *Tap42* gene can induce T-2 synthesis of *F. oxysporum*. Targeted gene deletion has a significant effect on fungal metabolism, and sometimes will specifically activate other metabolism pathways. He et al. [1] reported that inactivation of TORC1-Tap42 leads to autophagy in yeast. The addition of neutral AAs, such as L-Ile, L-Leu and L-Ser, increased T-2 production of Fo17, with values reaching up to 28.49, 27.07 and 29.61 ng/mL, respectively. Similarly, L-Phe, L-Pro and L-Cys significantly activated (*p* < 0.05) the T-2 production in Fo17, with values up to 30.49, 30.95 and 33.64 ng/mL, respectively (*p* < 0.05). However, acidic AAs L-Asp and L-Glu, as N sources, significantly decreased (*p* < 0.05) the T-2 toxin-production in Fo17 to 14.87 and 12.64 ng/mL, respectively (*p* < 0.05). After *Tap42* deletion, the T-2 production was significantly decreased (*p* < 0.05) to 6.68 and 2.09 ng/mL. Thus, acidic AAs were not conducive to *F. oxysporum* growth and secondary metabolism, and sporulation was the main path to survival. However, the inhibition of toxin production does not mean toxicity reduction. When pathogenic fungi are exposed to pH stress, AAs are used for biotransformation and to drive environmental alkalinization for growth. *Candida albicans* uses AAs as the sole N source to raise the environmental pH from 4 to neutral within 6–8 h, which is another critical pathogenicity trait [41].

Compared with Fo17, the T-2-producing capacity of ΔFo*Tap42* was significantly increased (*p* < 0.05) by L-His and L-Val addition, which indicates that *Tap42* negatively regulates L-His and L-Val absorption. The L-Asp and L-Glu addition significantly inhibited the T-2 synthesis by ΔFo*Tap42* to only 6.68 and 2.09 ng T-2/mL, respectively (*p* < 0.05). L-Ala, L-Arg, L-Cys, Gly, L-Leu, L-Pro, L-Ser, L-Thr and L-Try addition as N sources also reduced the T-2 production by ΔFo*Tap42*, which may be due to a lack of AA absorption and metabolism by the corresponding pathway gene (*Tap42*), resulting in a decline in T-2 production. A similar result was reported by Phasha et al. [42], who similarly demonstrated that the *Ras2* gene can control growth and toxin synthesis of *F. circinatum* though encoding mitogen-activated protein kinase.

As shown in Figure 2, heat map analysis showed the effect patterns of AAs on sporulation and T-2 production by Fo17, ΔFo*Tap42* and ΔFo*Tap42*-C. During sporulation, L-Ile and L-Pro groups clustered closely together, showing significantly activated (*p* < 0.05) spore production of ΔFo*Tap42*, but a reduction in ΔFo*Tap42*-C, which indicates that *Tap42* negatively regulated L-Ile and L-Pro absorption and, thereby, affected the *F. oxysporum* spore formation. In contrast, in the L-His and L-Ser groups, the spore production of ΔFo*Tap42* was lower than in Fo17 and ΔFo*Tap42*-C, meaning that *Tap42* positively activated L-His and L-Ser absorption in *F. oxysporum*. When a lack of a regulated gene markedly reduces spore production, it indicates a decrease in reproductive dispersal capacity. The L-Ala group clustered with the control group means that L-Ala addition did not significantly affect the spore production of Fo17, ΔFo*Tap42* or ΔFo*Tap42*-C (Figure 2A). L-Asp and L-Glu addition inhibited T-2 production by Fo17, and a lack of *Tap42* significantly inhibited the T-2 production (<20 ng/mL). This is because acidic AAs negatively regulated the pathogenicity of *Fusarium* spp.. The L-His, L-Val, L-Lys and L-Met groups clustered together, and the T-2 production was significantly increased when *Tap42* was lacking (Figure 2B).

### 3.3. Dose-Effect Relationship between L-Thr, L-His and L-Asp on T-2 Toxin Synthesis by F. oxysporum Regulated by Tap42

*Tri5* is the initiating gene of trichothecene synthesis, and plays an important role in *Fusarium* spp. pathogenicity [43]. In this study, neutral (L-Thr), basic (L-His) and acidic (L-Asp) AAs were selected to assess the effect on T-2 synthesis and *Tri5* expression. Increasing L-Thr in the GYM medium significantly activated the T-2 production of Fo17, ΔFo*Tap42* and ΔFo*Tap42*-C (*p* < 0.05) with the concentrations of T-2 at 70 mg/mL L-Thr, 13.86, 16.87 and 13.85 ng/mL, respectively. Compared with the control group, a significant increase (*p* < 0.05) in *Tri5*’s expression of Fo17 was observed in the 50–70 mg/mL L-Thr groups. ΔFo*Tap42* showed a marked activation of *Tri5* expression in the 10–70 mg/mL L-Thr groups. When complemented with the *Tap42* gene, the *Tri5* expression was significantly increased (*p* < 0.05), but only in the 70 mg/mL L-Thr group (Figure 3A). Similarly, 10–70 mg/mL L-His addition significantly increased (*p* < 0.05) the T-2 production of Fo17, ΔFo*Tap42* and ΔFo*Tap42*-C, with the T-2 concentrations reaching 33.03, 31.66 and 32.75 ng/mL, respectively, in the 70 mg/mL L-His added group. However, the *Tri5* expression of Fo17 increased, but the increase in amplitude of ΔFo*Tap42* and ΔFo*Tap42*-C were lower than in Fo17 (Figure 3B). In the L-His group, higher T-2 production and lower *Tri5* expression were observed in ΔFo*Tap42*, which may due to the lack of the TORC1-*Tap42* gene, thus reducing the sensitivity to L-His and, therefore, reducing *Tri5* signal transduction. However, T-2 synthesis is not only regulated by *Tri5.* Multiple trichothecene synthetic gene clusters, such as three P450 oxygenase genes, *Tri4*, *Tri11* and *Tri13* [44]; esterase gene *Tri8* [45]; and acetyltransferase genes *Tri3* and *Tri7* [46] also affect T-2 synthesis. Conversely, 10 mg/mL L-Asp addition significantly increased (*p* < 0.05) T-2 production and *Tri5* expression, while L-Asp addition gradually reduced T-2 production and *Tri5* expression in Fo17, ΔFo*Tap42* and ΔFo*Tap42*-C, which indicated that a low dosage of L-Asp led to a significant increase in the growth and T-2 production of *F. oxysporum*, whereas a high dosage reduced this effect. An inhibitory effect of *Tri5* expression was detected in ΔFo*Tap42* following the addition of 70 mg/mL L-Asp. Thus, acidic AA L-Asp showed a significant reduction in *Fusarium* sp. growth and metabolism at a relatively high dose (Figure 3C). The inhibitory effect of L-Asp has been well-studied and applied for fungal inhibition in food products. Bitu et al. [47] used L-Asp combined with Th (IV) and Zr (IV) ions to synthesize new peroxo complexes, and demonstrated high antibacterial activity in *Aspergillusflavus*, *Penicillium* sp., *Candida* sp. and *Aspergillus niger*. Thus, based on its excellent antifungal properties, L-Asp can be used as an additive in the development of natural preservatives.

### 3.4. L-Thr, L-His and L-Asp Activated Tap42 Interacting Proteins and Metabolic Pathways

A Tap42 expression vector was constructed using BamHi-XhoI restriction enzyme digestion. Two bands (3359 bp and 5192 bp) were observed in the pET-28a (+) plasmid vector after PCR amplification, indicating that the target gene, *Tap42*, was cloned into the expression vector (Figure 4A). Through comparative analysis, the pET28a(+)-Tap42 expression vector was found to be ~140 kDa. *E. oil* was transformed with the correctly identified positive clone plasmid, and single bacterial colonies were selected for induction expression. The results of SDS-PAGE electrophoresis are shown in Figure 4B. The recombinant strain induced by IPTG showed a fusion protein, pET28a(+)-Tap42, with a target band of ~140 kDa (electrophoretic band 2–4), but not the non-induced strain (electrophoretic band 1). A purified protein (~140 kDa) was obtained after washing and eluting the protein (electrophoretic band 3) (Figure 4C). Based on SDS-PAGE, no significant interacting proteins of Fo17 were detected in the GYM medium (electrophoretic band 1). In the L-His group, clear protein bands were detected at ~10 and 39–43 kDa (electrophoretic band 2). In the L-Thr group, as well, clear protein bands were evident at ~10, 27 and 39–52 kDa (electrophoretic band 3). In the L-Asp group, bands were found at ~10, 27 and 39–52 kDa (electrophoretic band 4) (Figure 4D).

By Q Exactive LC-MS/MS analysis, 172, 201 and 89 potential interacting proteins of Tap42 were detected in the L-Thr, L-His and L-Asp treatment groups, respectively. A total of 86 interacting proteins co-existed in the three treatment groups. In the L-Thr and L-His groups, 84 similar interacting proteins were detected. However, only 1 and 2 interacting proteins from the L-Thr and L-His groups were similar to those observed in the L-Asp group (Figure 5). Based on mass analysis, the common potential interacting proteins with Tap42 treatment after L-Thr, L-His and L-Asp treatments were mostly DNA topoisomerase 2, glyceraldehyde-3-phosphate dehydrogenase 2, heat shock 70 kDa protein, elongation factor 1-alpha, actin, plasma membrane ATPase, enolase, transaldolase and ATP-dependent RNA helicase and citrate synthase (Table 2). Comparative analysis showed that the L-His and L-Asp groups specifically activated the ribosome-associated molecular chaperone SSB1, regulated by *SSB1,* and phosphoglycerate kinase regulated by *pgkA,* respectively. In the L-Thr and L-His groups, pyruvate decarboxylase regulated by *pdcA* was markedly activated. L-His specifically activated the Tubulin alpha-B chain regulated by *tba-2* (Table 3). DNA topoisomerases were the enzymes mostly responsible for the encapsulation of double-helix DNA and the transcription of proteins, and were shown to play an important role in the growth and metabolism of fungi [48]. *Candida albicans* and *Aspergillus niger* have high levels of both type I and type II topoisomerases, and increase pathogenicity by converting cellular proteins [49]. Therefore, the inhibition of DNA topoisomerases activity is also an important target of antifungal drugs [50]. Glyceraldehyde 3-phosphate dehydrogenase is an enzymatic protein highly related to the catalyzation of oxidation (dehydrogenation) and phosphorylation of glyceraldehyde 3-phosphate to generate 1, 3-diphosphate glyceric acid, which is the central link to the glycolytic pathway and, hence, plays an important role in glycometabolism [43]. In addition, Tap42-related enolase and pyruvate decarboxylase are important enzymes in the glycolytic pathway, and play important roles in maintaining cell wall protection and fungal pathogenicity [51]. The interaction of Tap42 with these enzymatic proteins cultured in GYM medium indicates that the growth and metabolism of *Fusarium* firstly utilize glucose through the glycolytic pathway to acquire energy.

According to KEGG enrichment analysis, Tap42-interacting proteins were mostly derived from metabolism, genetic information processing, cellular processes and environmental information processing. Most of the interacting proteins came from the metabolic pathways, mainly from glycerophospholipid metabolism, pentose phosphate pathway, glyoxylate and dicarboxylate metabolism, glycolysis, gluconeogenesis, methane metabolism and glutathione metabolism. The related secondary metabolic pathways were galactose metabolism, fatty acid biosynthesis, fatty acid degradation, steroid biosynthesis, phenylalanine metabolism, one carbon pool by folate, citrate cycle, TCA cycle and oxidative phosphorylation (Figure 6).

KEGG enrichment analysis showed that the proteins potentially interacting with Tap42 were from the mitogen-activated protein kinase (MAPK) signaling pathway and Hippo signaling pathway. The MAPK signaling pathway is composed of mitogen-activated protein kinase, which is involved in the intracellular signal regulation system and can be activated by different extracellular stimuli, such as cytokine neurotransmitter hormones, cell stress and cell adhesion [52,53]. Many reports have shown that the MAPK pathway is significantly associated with the regulation of the TORC1 pathway [54]. The Hippo signaling pathway mainly inhibits cell growth and regulates cell proliferation, apoptosis and stem cell self-renewal [55]. Recent studies have shown that the Hippo signaling pathway is closely related to cancer genesis, tissue regeneration and stem cell function regulation [56]. Identification of the associated signaling pathway provides a theoretical basis for the further exploration of the interaction network of *Fusarium* pathogenicity when exposed to different amino acid conditions, and also provides important information for the further control of fungal contamination in high-protein fermented foods.

## 4. Conclusions

Exposure to free AAs (as N sources) showed a significant effect on the growth and metabolism of *F. oxysporum* regulated by *Tap42*. The absorption of L-Ile and L-Tyr was regulated by *Tap42*. Acidic (L-Asp, L-Glu) and S-containing (L-Cys, L-Met) AAs were not conducive to *F. oxysporum* growth, and were not regulated by *Tap42*. The addition of L-Ile and L-Pro activated sporulation in ΔFo*Tap42,* but this was negatively regulated by *Tap42*, while L-His and L-Ser inhibited sporulation in ΔFo*Tap42,* which was positively regulated by *Tap42*. Acidic AA showed a remarkable inhibitory effect on T-2 toxin production positively regulated by *Tap42*. Neutral (L-Thr) and alkaline (L-His) AAs significantly activated the T-2 synthesis and *Tri5* expression of *Fusarium,* while L-Asp showed an inhibitory effect at relatively high doses. The co-immunoprecipitation analysis showed that the interacting proteins of Tap42 were activated by L-The, L-His and L-Asp control metabolism; genetic information processing; cellular processes and environmental information processing. L-Asp inhibited the effects on *Fusarium* spp. growth and metabolism, and, thus, could be used as an inhibitor to further control fungal contamination in fermented foods.

## Figures and Tables

**Figure 1 foods-12-01829-f001:**
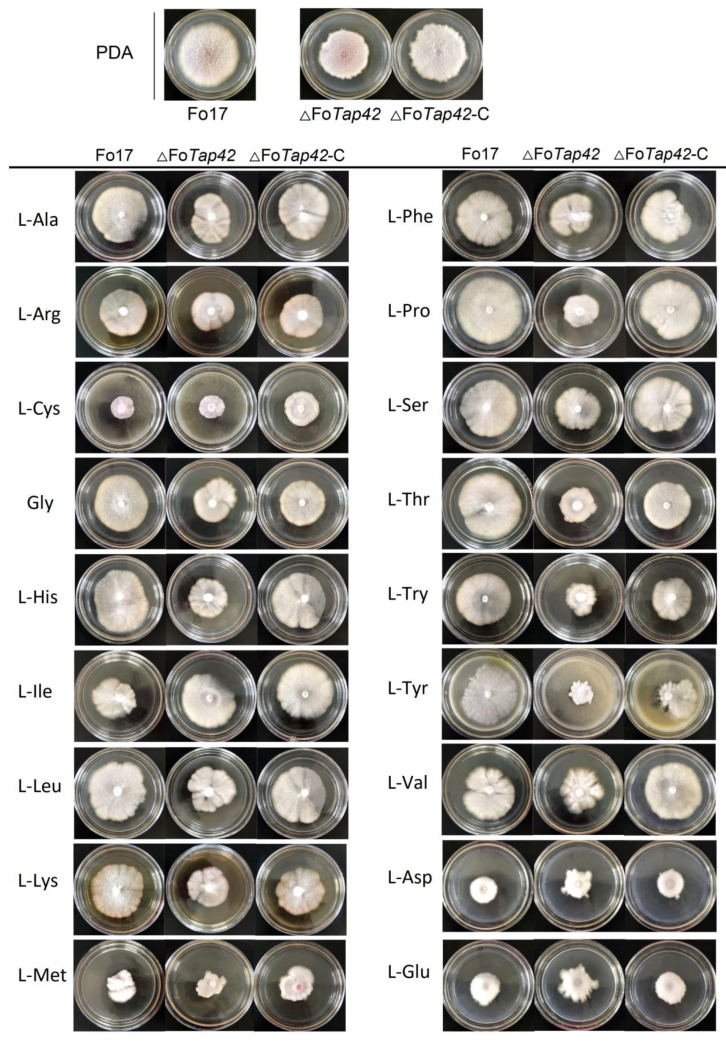
Colony morphology of wild-type *F. oxysporum* Fo17, ΔFo*Tap42* and ΔFo*Tap42*-C in dextrose agar medium, supplemented with 18 amino acids cultured for 7 d at 28 °C (*n* = 3). Control was cultured in the PDA medium.

**Figure 2 foods-12-01829-f002:**
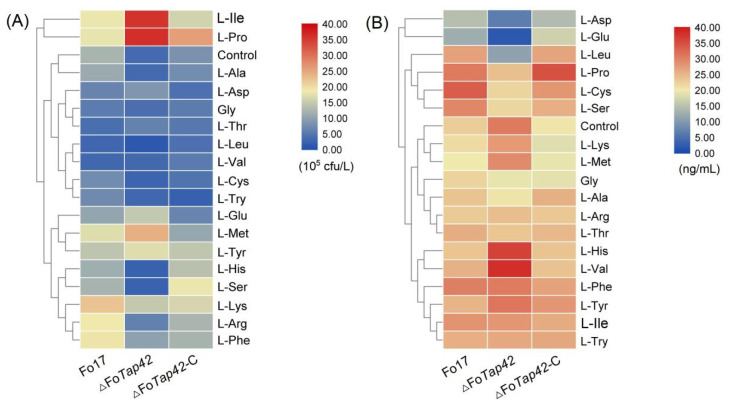
Changed patterns of sporation (**A**) and T-2 synthesis (**B**) by *F. oxysporum* Fo17, ΔFo*Tap42* and ΔFo*Tap42*-C exposed to 18 amino acids as nitrogen sources (*n* = 3). Control was cultured in CDA and GYM media.

**Figure 3 foods-12-01829-f003:**
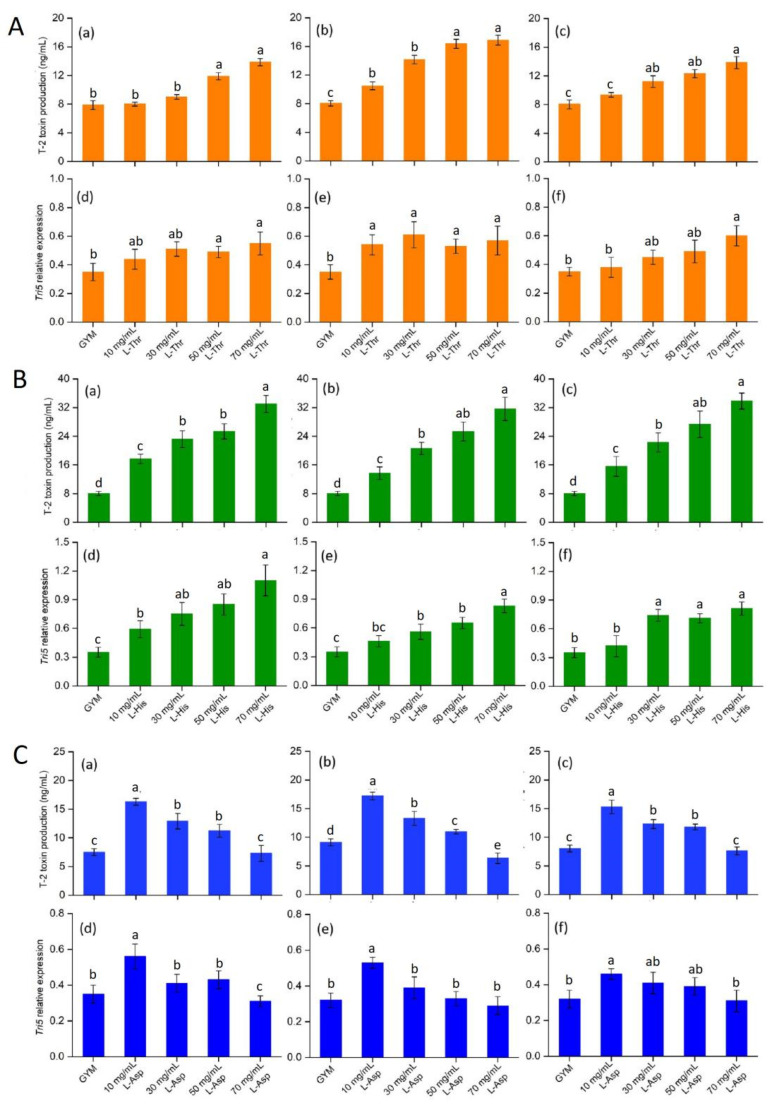
Effects of L-Threonine (L-Thr) (**A**), L-Histidine (L-His) (**B**) and L-Aspartic acid (L-Asp) (**C**) on T-2 toxin production and *Tri5* expression of different strains (*n* = 3), cultured at 28 °C for 14 d. The control was cultured in GYM medium. (a)–(c): T-2 toxin production of wild-type Fo17, ΔFo*Tap42* and ΔFo*Tap42*-C; (d)–(f): *Tri5* expression of wild-type Fo17, ΔFo*Tap42* and ΔFo*Tap42*-C. Different letters “a–d” in the same plot indicate significant differences (*p* < 0.05).

**Figure 4 foods-12-01829-f004:**
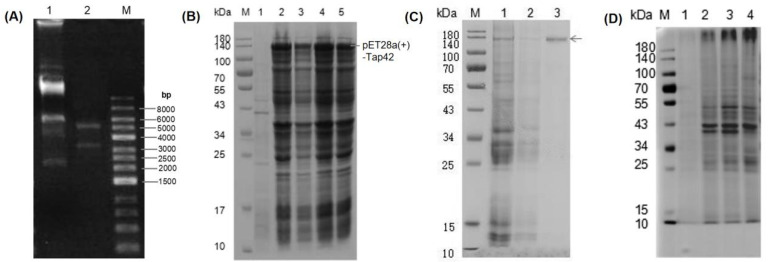
(**A**) PCR amplification analysis of a plasmid digested by the XbaI-XhoI enzyme. A1: plasmid; A2: XbaI-XhoI digestion plasmid; M: DNA marker. (**B**) Prokaryotic expression of pET28a(+)-Tap42 fusion protein. B1: protein not induced by IPTG, B2-B4: clone 1–3 protein induced by IPTG. (**C**) Purification of pET28a(+)-Tap42 fusion protein. M: marker; C1: unpurified protein; C2: washed protein; C3: eluted protein; M: molecular protein marker. (**D**) Co-immunoprecipitation products of Tap42 protein identified by SDS-PAGE. D1: GYM control group, D2: L-Histidine (L-His)-treated group, D3: L-Threonine (L-Thr)-treated group; D4: L-Aspartic acid (L-Asp)-treated group, M: molecular protein marker.

**Figure 5 foods-12-01829-f005:**
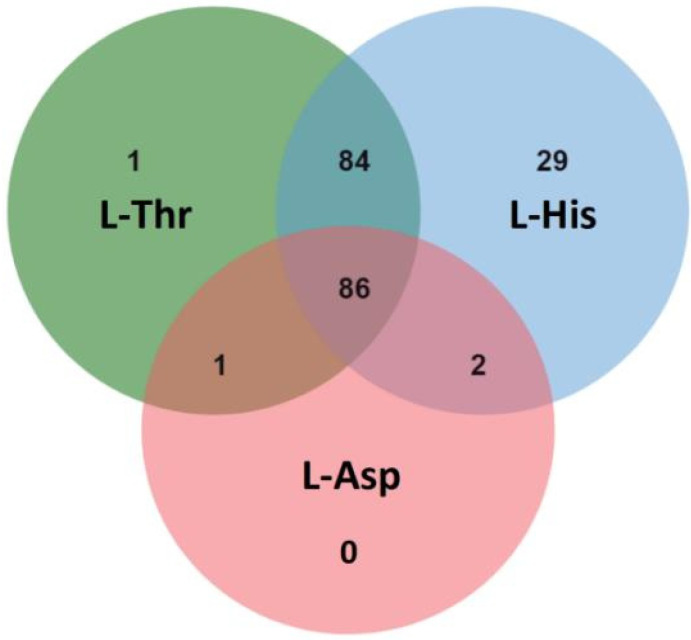
The common and specific numbers of potential Tap42-interacting proteins in *F. oxysporum* following L-threonine (L-Thr), L-Histidine (L-His) and L-Aspartic acid (L-Asp) treatments (*n* = 3).

**Figure 6 foods-12-01829-f006:**
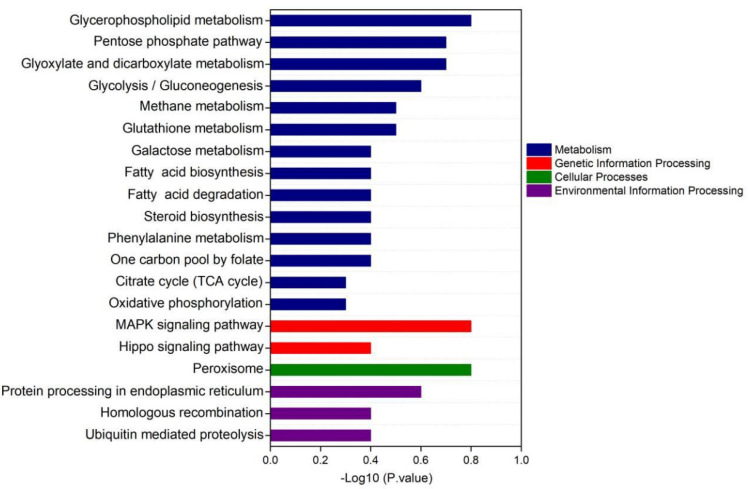
The KEGG enrichment analysis of the signaling pathway of Tap42, potentially interacting with proteins of *F. oxysporum* Fo17 cultured with L−threonine (L−Thr), L−Histidine (L−His) and L−Aspartic acid (L−Asp) (*n* = 3).

**Table 1 foods-12-01829-t001:** Effects of 18 amino acids as nitrogen sources in T-2 toxin and spore production by *F. oxysporum* Fo17, ΔFo*Tap42* and ΔFo*Tap42*-C.

Group	Spore Concentration (10^5^ CFU/L)			T-2 Toxin Concentration (ng/mL)		
	Fo17	ΔFo*Tap42*	ΔFo*Tap42*-C	Fo17	ΔFo*Tap42*	ΔFo*Tap42*-C
Control	12.35 ± 1.32 ^c^	3.51 ± 1.12 ^e^*	8.24 ± 1.62 ^bc^*	22.86 ± 3.54 ^bc^	30.82 ± 3.41 ^b^*	20.69 ± 1.82 ^bc^
L-Ala	11.11 ± 2.25 ^c^	3.31 ± 0.76 ^e^*	7.61 ± 1.24 ^bc^*	23.85 ± 4.34 ^bc^	20.73 ± 4.61 ^c^	25.63 ± 4.41 ^b^
L-Asp	6.52 ± 0.64 ^d^	8.75 ± 3.83 ^de^	4.25 ± 0.71 ^c^	14.87 ± 2.94 ^c^	6.68 ± 2.65 ^e^*	14.43 ± 3.66 ^d^
L-Arg	18.80 ± 1.35 ^b^	6.25 ± 1.70 ^de^*	12.50 ± 1.13 ^ab^*	23.19 ± 4.01 ^bc^	24.32 ± 3.83 ^c^	23.39 ± 3.64 ^b^
L-Cys	7.50 ± 1.32 ^d^	2.75 ± 1.11 ^e^*	4.62 ± 0.92 ^c^*	33.64 ± 1.41 ^a^	22.23 ± 2.08 ^c^*	28.21 ± 4.36 ^ab^
L-Glu	10.52 ± 2.10 ^c^	14.80 ± 4.62 ^c^*	6.75 ± 0.35 ^c^*	12.64 ± 1.40 ^c^	2.09 ± 1.61 ^e^*	17.03 ± 5.27 ^c^
Gly	5.56 ± 0.61 ^de^	4.00 ± 1.23 ^de^	5.64 ± 0.95 ^c^	22.39 ± 0.36 ^bc^	19.43 ± 4.05 ^c^	19.08 ± 2.03 ^c^
L-His	11.59 ± 2.46 ^c^	2.50 ± 0.64 ^e^*	13.88 ± 1.66 ^ab^	23.67 ± 3.72 ^bc^	36.38 ± 0.64 ^a^*	23.87 ± 4.08 ^b^
L-Ile	18.30 ± 3.65 ^b^	35.80 ± 4.87 ^a^*	16.05 ± 1.45 ^a^	28.49 ± 1.32 ^ab^	28.03 ± 5.46 ^b^	26.15 ± 4.47 ^ab^
L-Leu	3.02 ± 0.57 ^e^	1.25 ± 0.36 ^e^*	4.25 ± 1.04 ^c^	27.07 ± 2.25 ^ab^	11.22 ± 1.54 ^d^*	26.81 ± 1.44 ^ab^
L-Lys	22.30 ± 3.55 ^a^	14.85 ± 0.98 ^c^*	16.32 ± 3.52 ^a^*	21.51 ± 2.77 ^bc^	27.78 ± 1.39 ^bc^	18.71 ± 2.92 ^c^
L-Met	17.20 ± 1.28 ^b^	24.21 ± 1.16 ^b^*	10.84 ± 2.69 ^b^*	20.19 ± 1.54 ^bc^	29.25 ± 2.02 ^b^*	19.43 ± 3.91 ^c^
L-Phe	19.34 ± 13.35 ^b^	10.01 ± 2.06 ^d^*	14.30 ± 0.98 ^a^*	30.49 ± 1.54 ^a^	30.84 ± 3.05 ^b^	26.95 ± 2.98 ^ab^
L-Pro	18.05 ± 4.21 ^b^	36.26 ± 4.53 ^a^*	15.80 ± 3.36 ^a^	30.95 ± 5.17 ^a^	24.29 ± 5.37 ^c^	34.78 ± 4.19 ^a^
L-Ser	12.24 ± 2.17 ^c^	2.56 ± 0.53 ^e^*	14.32 ± 4.39 ^a^	29.61 ± 2.97 ^a^	22.02 ± 0.64 ^c^	25.79 ± 4.33 ^b^
L-Thr	4.01 ± 0.75 ^e^	6.32 ± 1.27 ^de^	5.11 ± 1.24 ^cd^	25.96 ± 1.54 ^b^	23.58 ± 1.36 ^c^	24.85 ± 1.57 ^b^
L-Try	7.25 ± 0.69 ^d^	3.21 ± 0.28 ^e^*	6.25 ± 0.35 ^c^	25.75 ± 1.61 ^b^	26.41 ± 2.62 ^bc^	26.56 ± 5.17 ^ab^
L-Tyr	14.30 ± 3.22 ^c^	17.25 ± 3.54 ^c^*	14.5 ± 2.12 ^a^	25.26 ± 1.88 ^b^	31.22 ± 4.92 ^b^*	28.03 ± 1.68 ^ab^
L-Val	3.14 ± 0.53 ^e^	5.25 ± 1.20 ^e^	4.51 ± 1.31 ^d^	25.54 ± 3.91 ^b^	38.15 ± 0.64 ^a^*	23.93 ± 3.02 ^b^

Three strains were cultured in CDA and GYM media (nitrogen removal sources) and supplemented with 18 amino acids for 14 d (*n* = 3). The original CDA and GYM media were the controls. Different letters (a–e) in the same column indicate significant differences (*p* < 0.05). Compared with Fo17, significant differences (*p* < 0.05) of ΔFo*Tap42* and ΔFo*Tap42*-C are marked with ‘*’ in each row.

**Table 2 foods-12-01829-t002:** Common potential interacting proteins of the Tap42 immunoprecipitation products treated by L-Thr, L-His and L-Asp, identified by mass spectrometry.

Accession	Gene Name	Protein Name	Molecular Mass	Score
O93794	TOP2	DNA topoisomerase 2	161,402	1039
P17730	gpd2	Glyceraldehyde-3-phosphate dehydrogenase 2	36,198	904
P54117	GPDA	Glyceraldehyde-3-phosphate dehydrogenase	36,405	795
P06786	TOP2	DNA topoisomerase	164,626	784
P35143	GPDA	Glyceraldehyde-3-phosphate dehydrogenase	36,500	712
Q01233	hsps-1	Heat shock 70 kDa protein	70,738	683
Q01372	tef-1	Elongation factor 1-alpha	49,983	612
Q6TCF2	ACT	Actin	41,809	604
Q07421	PMA1	Plasma membrane ATPase	99,393	554
Q6RG04	ENO1	Enolase	47,388	553
Q76KF9	enoA	Enolase	47,264	545
Q01765	TEF	Elongation factor 1-alpha	50,215	500
P23704	atp-2	ATP synthase subunit beta	55,556	464
A0A075DVI9	FPRO05_10296	Transaldolase	35,571	389
P10592	SSA2	Heat shock protein SSA2	69,599	368
P37211	atp-1	ATP synthase subunit alpha	59,713	318
P07038	pma-1	Plasma membrane ATPase	100,280	299
Q96X45	cot-3	Elongation factor 2	93,545	270
A7EGL7	tif1	ATP-dependent RNA helicase eIF4A	45,088	237
Q8X097	B14D6.310	ATP-citrate synthase subunit 1	73,037	201
P28876	pma2	Plasma membrane ATPase 2	110,743	199
Q99002	BMH1	14-3-3 protein homolog	30,094	195
C7YTD6	RPS1	40S ribosomal protein S1	29,288	185
Q99170	KAR2	reticulum chaperone BiP	73,593	178
P34085	cit-1	Citrate synthase	52,241	166
C7C436	mcsA	2-methylcitrate synthase	52,184	159
P50142	HSP60	Heat shock protein 60	62,079	158
P0C016	ubi3	Ubiquitin-40S ribosomal protein S27a	17,475	142
Q9P3A7	cdc48	Cell division cycle protein 48	90,354	139
Q5I2J3	TUB1	Tubulin alpha chain	50,737	137
P51044	cit-1	Citrate synthase	52,406	135
P24634	tubB	Tubulin alpha-2 chain	50,541	126
Q8J0N6	FBA2	Fructose-bisphosphate aldolase 2	39,973	126
Q5AWS6	cdc48	Cell division control protein 48	90,769	124
O13639	pi047	Adenosylhomocysteinase	47,866	121
Q7RVA8	ace-8	Pyruvate kinase	58,290	115
P49382	AAC	ADP, ATP carrier protein	33,187	111
Q12629	PDC1	Pyruvate decarboxylase	61,905	110
Q6C1F3	ENO	Enolase	47,277	108
Q9HES8	pyc	Pyruvate carboxylase	131,526	106
P87252	hex-1	Woronin body major protein	19,229	104
Q7RVI1	rps-5	40S ribosomal protein S5	23,836	103

The Tap42 potential interacting proteins of *F. oxysporum,* cultured in GYM medium, with L-threonine (L-Thr), L-Histidine (L-His) and L-Aspartic acid (L-Asp) (*n* = 3). A high score indicates that the protein is closely related to Tap42.

**Table 3 foods-12-01829-t003:** Proteins of *F. oxysporum* potentially interacting with Tap42, activated by specific amino acids.

Amino Acid Treatment	Accession	Gene Name	Protein Name	Molecular Mass	Score
L-His group L-Asp group	P11484	*SSB1*	Ribosome-associated molecular chaperone SSB1	66,732	234
	P41756	*pgkA*	Phosphoglycerate kinase	44,413	209
L-Thr group L-His group	Q0CNV1	*pdcA*	Pyruvate decarboxylase	63,206	111
L-His group	P38669	*tba-2*	Tubulin alpha-B chain	50,675	72

The Tap42 potential interacting proteins of *F. oxysporum* cultured in GYM medium with L-threonine (L-Thr), L-Histidine (L-His) and L-Aspartic acid (L-Asp) addition (*n* = 3). A high score indicates that the protein is closely related to Tap42.

## Data Availability

All related data and methods are presented in this paper. Additional inquiries should be addressed to the corresponding author.

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
