# Peer review of "Effect of Amino Acids on Fusarium oxysporum Growth and Pathogenicity Regulated by TORC1-Tap42 Gene and Related Interaction Protein Analysis"

_foods, 2023, doi:10.3390/foods12091829_

Round 1

Reviewer 1 Report

Dear authors.

Your paper is novelty and has valuable information, however, is complicated to have a full comprehension of the data without having the full information of the methodology section and the proper representation of data analyses in the tables.

I suggest some changes:

1.      Make a proper review of the basic concepts. The yeasts do not sporulate.

2.      Make the corrections concerning the format such as double and missing spaces.

3.      Review carefully all grammatical mistakes.

4.      Improve the EN language.

5.      Some methodological information is missing, such as in Section 2.6.4.

6.      Mention clearly the conditions used to grow the control.

7.      Use the letter system to demonstrate significant differences.

8.      Add the table description, a table should provide all the information to understand the information.

9.      Provide a full description of the figures including tested microorganism, growth conditions…

10.   The results section is well-written, but the methodological section must be improved.

Use a lower size letter to L and D.

Review all the missing and double spaces.

Improve the redaction of the methodology section.

CFU with capital letters.

Saccharomyces cerevisiae is a yeast, that does not produce spores.

Use the letter system to indicate significant differences.

Add the description to tables.

Review all the scientific names.

Without a clear explanation of methodology and statistical analyses is complicated to understand your results. Please, all the authors read carefully the MS.

Add the information on the controls. 

Author Response

Reviewer1:

Your paper is novelty and has valuable information, however, is complicated to have a full comprehension of the data without having the full information of the methodology section and the proper representation of data analyses in the tables.

I suggest some changes:

  1. Make a proper review of the basic concepts. The yeasts do not sporulate.

Answer: Thanks for the reminding. Saccharomyces cerevisiae reproduce by budding in normal lifereproduction and does not produce spores. However, under harsh environmental conditions, they can reproduce by meiosis and produce haploid spores. In Part 3.2, an article was quoted which reported that TORC1 can activated Saccharomyces cerevisiae to produce spores under the glucose-dependent condition. We quote this paper in order to prove that TOCR1-Tap42 had a regulatory effect on regulating expression of glucose-response genes and spore formation. Also we have changed this sentence to make it more clear.

2.Make the corrections concerning the format such as double and missing spaces.

Answer: We have corrected the format and the double and missing spaces in this manuscript.

3.Review carefully all grammatical mistakes.

Answer: We have checked and changed all grammatical mistakes.

4.Improve the EN language.

Answer: We have improved the English grammatical expression.

  1. Some methodological information is missing, such as in Section 2.6.4.

Answer: Thanks for the suggestion. We have added the method information in this part.

6.Mention clearly the conditions used to grow the control.

Answer: Thanks for the suggestion. We have added the control information in each experiment part.

7.Use the letter system to demonstrate significant differences.

Answer: Thanks for the suggestion. We have changed the significant differences to letter system and modified description of this part.

  1. Add the table description, a table should provide all the information to understand the information.

Answer: Thanks for the suggestion. We have added more information in all tables.

9.Provide a full description of the figures including tested microorganism, growth conditions…

Answer: Thanks for the suggestion. We have added more information in all figures.

  1. The results section is well-written, but themethodological sectionmust be improved.

Answer: We have corrected the methodological section and added more information in this part.

Use a lower size letter to L and D.

Answer: We have changed L and D to lower size letter.

Review all the missing and double spaces.

Answer: We have checked all the missing and double spaces and changed it.

Improve the redaction of the methodology section.

Answer: We have improved the description of experimental method.

CFU with capital letters.

Answer: We have change all cfu to CFU.

Use the letter system to indicate significant differences.

Answer: Thanks for the suggestion. We have changed the significant differences to letter system and modified description of this part.

Add the description to tables.

Answer: Thanks for the suggestion. We have added the detail description in each tables.

Review all the scientific names.

Answer: Thanks for the suggestion. We have checked the scientific names and corrected it.

Without a clear explanation of methodology and statistical analyses is complicated to understand your results. Please, all the authors read carefully the MS.

Answer: Thanks for the suggestion. We have changed the statistical analyses to intergroup analysis and use the letter system to demonstrate significant differences.

Add the information on the controls.

Answer: Thanks for the suggestion. We have added the control information in all tables and figures.

Reviewer 2 Report

A very interesting manuscript on the effect of amino acids on Fusarium oxysporum growth and pathogenicity and the possible role of TORC1-Tap42.

A few amendments are necessary:

1.   paragraph 2.4. please ensure that numbers are subscripted (lines 1-3) or superscripted (l. 6, 9)

2.    paragraph 2.5. It should read ‘Tubulin’. In addition, primer sequences and qPCR conditions (composition of reaction mixture & thermocycling conditions) should be reported.

3.    paragraph 2.6.1. It should read ‘E. coli’ and ‘Rossetta’

4.    paragraph 2.6.4. How was SDS-PAGE performed?

5.    paragraph 3.2 and throughout the text. CFU is an acronym and should be written with capitalized letters.

6.    table 1. ANOVA per rows and columns should be reported in the table with superscript letters.

7.    the results presented in paragraph 3.3. are not consistent with the results shown in fig. 3. For example, elevation and activation of Tri5 expression is claimed in the text but in fig. 3 a clear downregulation is shown.   

8.    is fig 4A sds-page? it looks like agarose gel electrophoresis. what %acrylamide allows electrophoresis of 8 kb dna fragment?

Author Response

Reviewer2:

A very interesting manuscript on the effect of amino acids on Fusarium oxysporum growth and pathogenicity and the possible role of TORC1-Tap42.

A few amendments are necessary:

  1. paragraph 2.4. please ensure that numbers are subscripted (lines 1-3) or superscripted (l. 6, 9)

Answer: Thanks for the suggestion. We have changed it.

  1. paragraph 2.5. It should read ‘Tubulin’. In addition, primer sequences and qPCR conditions (composition of reaction mixture & thermocycling conditions) should be reported.

Answer: Thanks for the suggestion. We have added the primer sequences and qPCR conditions in part 2.5.

  1. paragraph 2.6.1. It should read ‘E. coli’ and ‘Rossetta’

Answer: Thanks for the suggestion. We have changed E. coli to E. coli and Rosstta to Rossetta.

 .

  1. paragraph 2.6.4. How was SDS-PAGEperformed?

Answer:We have added the SDS-PAGE method in this part.

5.paragraph 3.2 and throughout the text. CFU is an acronym and should be written with capitalized letters.

Answer: Thanks for the suggestion. We have change all cfu to CFU.

  1. table 1. ANOVA per rows and columns should be reported in the table with superscript letters.

Answer: Thanks for the suggestion. We have changed the significant differences to letter system and modified description of this part.

  1. the results presented in paragraph 3.3. are not consistent with the results shown in fig. 3. For example, elevation and activation of Tri5 expression is claimed in the text but in fig. 3 a clear downregulation is shown.   

Answer: Thanks for the suggestion. We have changed the results description in Fig.a.

  1. is fig 4A sds-page? it looks like agarose gel electrophoresis. what %acrylamide allows electrophoresis of 8 kb dna fragment?

Answer: It is the mistake. It should be PCR amplification analysis of plasmid digested by XbaI-XhoI enzyme. We have changed it.

Round 2

Reviewer 1 Report

Thank you for attending the comments

Author Response

Thank you.

Reviewer 2 Report

I am afraid the authors are not willing to improve the manuscript according to the recommendations. I will keep the numbering of the first revision to facilitate understanding.

2. paragraph 2.5. The primer sequences for Tubulin are still not presented, the composition of the reaction mixture is still not presented and the thermocycling conditions added are clearly not for qPCR (final extension step is not used in qPCR). I would also like to see if melting curve analysis was performed as verification step, which was also not included.

4. paragraph 2.6.4. the text added seems to be, at least, truncated. Was SDS-PAGE or denaturing SDS-PAGE performed? In the case of the latter, the type and concentration of the denaturing factors should have been reported. I am not willing to comment the reported polyacrylamide concentration of 6% and the stable pressure of 1000 V.

6. Table 1. The authors have performed statistical analysis per columns; what about per rows?

7. the results presented in paragraph 3.3. are still not consistent with the results shown in fig. 3.

Author Response

I am afraid the authors are not willing to improve the manuscript according to the recommendations. I will keep the numbering of the first revision to facilitate understanding.

Answer: Thank you for your suggestions. We are willing to improve the manuscript according to the recommendations.

  1. paragraph 2.5. The primer sequences for Tubulin are still not presented, the composition of the reaction mixture is still not presented and the thermocycling conditions added are clearly not for qPCR (final extension step is not used in qPCR). I would also like to see if melting curve analysis was performed as verification step, which was also not included.

Answer: We have added the information about the primer sequences for β-Tubulin, the composition of the reaction mixture and corrected the thermocycling conditions for qPCR in manuscript (in red).

We have conducted the dissociation assay, as shown in the figure below, all primer sets had a clear melting peak which mean the PCR products were specific. This description has been provided in the manuscript.

  1. paragraph 2.6.4. the text added seems to be, at least, truncated. Was SDS-PAGE or denaturing SDS-PAGE performed? In the case of the latter, the type and concentration of the denaturing factors should have been reported. I am not willing to comment the reported polyacrylamide concentration of 6% and the stable pressure of 1000 V.

Answer: In this study, we performed denaturing polyacrylamide gel electrophoresis. Brifely, polyacrylamide gels (6%) were prepared in Bio-Rad chambers (separation gel: 2.5 μL [acrylamide 30%, bisacrylamide 0.8%], 1.5 μL buffer Tris [1.5 M pH 8.8], 52.5 μL sodium dodecyl sulfate [SDS,10%], 955 μL distilled water, 150 μL ammonium persulfate [PSA], and 7.5 μL N,N,N′,N′-tetramethyl-ethylenediamine [TEMED]; stacking gel: 312 μL [acrylamide 30%, bisacrylamide 0.8%], 450 μL buffer Tris [0.5 M pH 6.8], 1.0 μL distilled water, 50 μL PSA, and 4.0 μL TEMED). The electrophoresis chamber was filled with running buffer (1.44% glycine, 0.3% Tris, and 0.1% SDS). Then, samples were mixed with one volume of load buffer (187.5 mM Tris [pH 6.8], 6% SDS, 30% glycerol, and 15% β-mercaptoethanol), before being incubated at 98 oC for 5 min to completely denature proteins. The sample solutions (10 μL) and 10 μL of protein marker were loaded onto the gel. Electrophoresis was performed at a constant voltage of 50 V until the sample reached the separation gel, where the voltage was increased to 100 V.The gel staining was performed with stained with Protein Stain Q Kit (Sangon Biotech, Shanghai, China). Gel images were processed using the Image lab Software version 6.01 from Bio-Rad, adjusting the gamma setting to improve the contrast.

We have also revised the method part “2.6.4 SDS-PAGE and silver-staining” in manuscript (in red).

  1. Table 1. The authors have performed statistical analysis per columns; what about per rows?

Answer: We have added a statistical analysis per rows and discussed in paragraph 3.2 in manuscript (in red).

  1. the results presented in paragraph 3.3. are still not consistent with the results shown in fig. 3.

Answer: We have corrected the description of paragraph 3.3 in manuscript (in red).
